# Impact of the COVID-19 Pandemic on Black Communities in Canada

**DOI:** 10.3390/ijerph20021580

**Published:** 2023-01-15

**Authors:** Janet Kemei, Mia Tulli, Adedoyin Olanlesi-Aliu, Modupe Tunde-Byass, Bukola Salami

**Affiliations:** 1Faculty of Nursing, University of Alberta, Edmonton, AB T6G 1C9, Canada; 2Department of Obstetrics and Gynaecology, University of Toronto, Toronto, ON M5S 1A1, Canada

**Keywords:** COVID-19, black communities, health inequities

## Abstract

The COVID-19 pandemic has resulted in differential impacts on the Black communities in Canada and has unmasked existing race-related health inequities. The purpose of this study was to illuminate the impacts of the COVID-19 pandemic on Black people in Canada. Historically, social inequalities have determined the impacts of pandemics on the population, and in the case of the COVID-19 pandemic, disproportionate infections and mortalities have become evident among racialized communities in Canada. This qualitative descriptive study utilized an intersectionality framework. We invited Black stakeholders across Canada to participate in semi-structured interviews to deepen our knowledge of the impacts of the COVID-19 pandemic on Black communities in Canada. A total of 30 interviews were recorded, transcribed verbatim, and analyzed using content analysis. Our findings fell into three categories: (1) increased vulnerability to COVID-19 disease, (2) mental impacts, and (3) addressing impacts of the COVID-19 pandemic. The findings show the underlying systemic inequities in Canada and systemic racism exacerbated health inequities among the Black communities and undermined interventions by public health agencies to curb the spread of COVID-19 and associated impacts on Black and other racialized communities. The paper concludes by identifying critical areas for future intervention in policy and practice.

## 1. Introduction

The consequences of the COVID-19 pandemic have been profound all over the world, to varying degrees attributed to disease burden or enormous interruption to people’s daily activities. While the pandemic has affected the entire world, the health risks, burdens, experiences, and outcomes were not equal for everyone [1]. Black communities were disproportionately affected, with higher rates of diagnosis, hospitalization, and deaths from COVID-19 than other racial groups [2]. Inconsistent race-based data are available for Canada, but what evidence exists suggests that the disproportionate impact of the COVID-19 pandemic on Black and other racialized communities in Canada is similar to that seen in the US and UK [3,4,5]. Specifically, neighborhoods in Canada with the most significant proportion of Black and racialized communities have higher rates of COVID-19 infection and death [6]. The Canadian Race Foundation [7] defines Black Canadians as people of African descent and those who define themselves as such who are now residents or citizens of Canada.

Complex issues, including structural inequalities, have been suggested to negatively affect Black people and other marginalized communities in Canada, increasing their vulnerability to poor health outcomes [8]. Historically, social inequalities determine the impacts of pandemics on the population [9,10], such as disproportionate economic hardships for lower-income people [11]. Similarly, the COVID-19 pandemic has resulted in disproportionate challenges among racialized communities in Canada [11]. These communities suffer from a significant lack of access to medical and mental health services, precarious and high-risk employment, and lack of access to information about the pandemic [12].

Moreover, a statement from Black health leaders on the impact of the COVID-19 pandemic on Black communities in Ontario indicated that anti-Black racism exposed Black people to disproportionately higher rates of poverty, poorer health outcomes, and overrepresentation in the criminal system [13]. Anti-Black racism is defined as policies and practices rooted in Canadian institutions such as education, healthcare, and justice that mirror and reinforce beliefs, attitudes, prejudice, stereotyping, and discrimination towards people of Black-African descent [14]. Anti-Black racism makes it harder for Black people to own a home, resulting in higher housing density and multigenerational households, which by design makes it harder to social distance and heightens the risk for the transmission of the virus [3]. For instance, Statistics Canada (2020) reported a high loss of job and unemployment rates (17.6%) among Blacks compared to their South Asian (16.6%), White (9.4%), and Chinese Canadian (13.2%) counterparts in Canada during the COVID-19 pandemic [15]. While some Black Canadians lost their jobs, some still worked as frontline workers or in low-paying jobs and/or lived in crowded housing, rendering them more vulnerable to exposure to COVID-19 [16]. The aggravated burden of unemployment may also have intensified Black Canadians’ willingness and need to accept jobs that amplified their risk of contracting COVID-19 [15,17,18]. Of particular note, anti-Black racism increases disinformation and misinformation about COVID-19 and vaccines, causes distrust in the healthcare system, and contributes to vaccine hesitancy and poor adherence to public health interventions among these communities [18,19]. Further, contextual factors such as communication could proliferate vaccine hesitancy [20].

Given the novelty of the COVID-19 pandemic and the scarcity of racially disaggregated data in Canada, the impacts on Black people in Canada have not been widely documented. Considering the disproportionate impact of the COVID-19 pandemic on Canada’s Black communities, understanding how the pandemic has affected existing social inequities is important in post-pandemic recovery processes. Hence, the aim of this paper was to illuminate stakeholder perspectives about the impact of the COVID-19 pandemic on Black people in Canada. We report part of the results from a larger project that aimed to generate evidence on COVID-19 online misinformation among Black Canadians. Research questions include: (1) What is the impact of COVID-19-related online disinformation on Black Canadians? (2) What are the perspectives of Black stakeholders on evidence-based practices to address COVID-19-related online disinformation among Black Canadians?

## 2. Materials and Methods

### 2.1. Study Design

Descriptive qualitative research methodologies [21] and Crenshaw’s intersectionality framework [22] guided this research. At the beginning of the study, we purposively invited and recruited seven Advisory Committee members comprising Black community leaders, Black medical doctors, and executive leaders of Black organizations in Canada. The Advisory Committee contributed to the development of a semi-structured interview outline that we used during our interviews with participants. The use of a qualitative descriptive methodology allowed us to comprehensively summarize participant experiences of the pandemic, including their concerns and responses to public health measures. The semi-structured interview outline allowed us to obtain information directly from the stakeholders without limiting their responses.

### 2.2. Participants and Recruitment

We used maximum variation purposive sampling [23] to recruit participants from a list of Black organizations in Canada. We also used snowballing sampling when we had difficulties recruiting participants. The participants included Black community leaders, leaders of Black-led organizations, and Black health service providers who had rich information about the impacts of COVID-19 in their communities.

### 2.3. Data Collection and Analysis

A member of the research team completed participant interviews between February and April 2022. All interviews were conducted in English via Zoom or on the telephone, as per the participant’s choice, with the option of French provided to francophones. The interviews lasted approximately 1 h and were audio-recorded and transcribed verbatim by a professional transcriptionist. The data collection and analysis process was iterative. Data coding and qualitative content analysis were completed using QSR International NVivo 12.6.1, software. Recognizing that Black communities are a heterogeneous group, we ensured representation from African immigrants, Caribbean immigrants, members of historic Black communities in Canada, and Blacks from the United States of America. We also considered issues related to age, gender, race, embedded inequalities, and intersecting influences during the analysis. We collected data on both gender and sex from all participants and did not restrict demographic questions on gender to binary conceptualizations. We disaggregated demographic data by age, gender/sex, place of origin, location in Canada, religion, immunization status, and role within the Black community. However, we did not find any variation in participant responses.

### 2.4. Ethics Approval

The University of Alberta Research Ethics Board approved our study (Pro00114392). We also obtained written consent from participants prior to commencing the interviews.

## 3. Results

A total of 30 Black Stakeholders from across six provinces in Canada participated, including three advisory committee members. The final number of participants was determined once data saturation was reached [23]. Sandelowski [24] confirms that this number is within the range of descriptive qualitative research. A total of 16 identified as male and 14 as female. Fifteen of the participants were from Alberta, 10 were from Ontario, two were from Nova Scotia, and one was from each British Colombia, Manitoba, and Saskatchewan. Two participants provided a range for their age. Participant sociodemographic characteristics are outlined in Table 1.

Our findings generated three categories related to the impact of the COVID-19 pandemic on Black people in Canada: (1) increased vulnerability to COVID-19 disease, (2) mental health impacts, and (3) addressing impacts of the COVID-19 pandemic.

### 3.1. Increased Vulnerability to COVID-19 Disease

Participants indicated that receiving and sharing COVID-19 disinformation among the Black communities in Canada at the beginning of the pandemic amplified Black people’s vulnerability to the effects of the COVID-19 pandemic by promoting vaccine hesitancy. They indicated that disinformation was perpetuated by the lack of timely and accurate information about COVID-19 and its vaccines, followed by mistrust in healthcare systems, socio-economic impacts, and systemic racism.

#### 3.1.1. Lack of Timely and Accurate Information about COVID-19 and Its Vaccines

Given the rapid emergence and complexity of the COVID-19 virus, public health agencies needed to execute measures to promptly contain it. However, the timely dissemination of accurate information to Black communities was lacking, causing individuals within these communities to fall victim to misinformation about the pandemic. For instance, the following participant explains:


*However, what I’m saying is in the absence of timely, ongoing, trusted and connected sources of information where people can go and just really understand the fullness of the virus or the vaccines, or what’s happening in the community. In the absence of those kinds of things, folks fill that void either with disinformation, assumptions, or general perceptions about the disease.*
(P003)

Further, although the public health agency of Canada updated the public with the required preventative measures as the pandemic situation evolved, the provinces also had their requirements, which sometimes did not match the federal agency requirements. Other participants highlighted how the inconsistent and disparate communication from public health agencies regarding prevention measures was a barrier to Black community adherence to these measures and vaccine uptake.


*It is [communication] a gap for the Black community because with all the confusing statements that have been released since, there are inconsistencies and a lot of holes in conversations. So, they do not trust them. If you do not trust, then why should you be vaccinated?*
(P019)

Another participant emphasized that communication during the pandemic was not culturally appropriate for Black communities in Canada. They highlighted how the information about public health measures was not tailored to Black communities, regardless of the historical mistrust of the healthcare system and the government.


*I’m not trying to polish all the people in the Black community with one brush. My only concern is that sometimes the way information is communicated is not culturally appropriate. I think we need to simplify information so that the information regarding public health can reach people of all levels, education levels. I found that sometimes the information that was being communicated related to COVID-19 was very scientific and to a level that only people who are well educated would understand that information. Also, we need to remember this is a population that has been bruised by racism.*
(P016)

This participant also underscored how language barriers, especially for immigrants, increased Black people’s inability to distinguish public health recommendations from related mis/disinformation, putting them at risk of poor choices regarding vaccine uptake. For example, the following participant explains.


*The other thing I was also concerned about is that not all Black communities speak English, so how could public health information reach out to those who don’t speak English as their first language?... rather than people receiving second-hand information from people that they know. Because I’m thinking if I’m living with my parents who don’t speak English, and I’m against the COVID-19 vaccination, I will come and tell them what I do understand about the COVID-19 vaccination, and I will convince them not to take the COVID-19 vaccination.*
(P016)

#### 3.1.2. Mistrust in Healthcare Systems

One participant indicated how negative historical and current encounters of Black people within the healthcare system in Canada decreased vaccine confidence among Black people.


*Although sometimes we need to also understand why Black people are hesitant to take the vaccine. Sometimes it is rooted in the fears they have about how the medical system—how the medical system has not favored them in the Western countries in the past… And the history of the past, where certain medication has been used on certain people, to Blacks, and then it affected them adversely.*
(P008)

Misinformation about the COVID-19 vaccine causing infertility was widely spread among the Black community. One participant denoted issues regarding infertility and the COVID-19 vaccine but stated this was mainly related to Black men who had negative encounters with the judicial system.


*Like I have noticed that all the males that like believe that it’s infertility or mind control, they’ve all had some kind of interaction with the incarceration system, whether they’ve been in prison, or they’re frustrated. I feel like it is those that already have a distrust or have had an unpleasant experience with public service or government in any way that are quick to believe the misconceptions.*
(P023)

Other participants explained that most Black people in Canada have strong ties with their country of origin. Participants described how the behaviors of Black people in Canada towards COVID-19 public health measures were influenced by the information shared with people in their countries of origin. For example, this participant explains.


*They kind of start questioning all these measures. In Africa, we don’t have healthcare which is as advanced as here. We don’t even have in place like all these stringent measures, and people are not dying. Some of them don’t even believe like people are dying here, right? So that’s how the situation back home in African countries somehow does influence their behavior towards the measures in place here.*
(P012)

#### 3.1.3. Socio-Economic Impacts

The participants explained that most people from Black communities in Canada worked in frontline jobs, increasing their possible exposure to COVID-19. However, isolation was difficult for those who lived in shared spaces with their families. The following participant expounds on this.


*Clearly, you know, a lot of the Black community, at least from here where I am, I see we are the ones who are directly on the ground, whether it’s work related to housekeeping or, like you say, healthcare work like nursing, direct care support, so you’re at the point of where you’re more likely to get infected because of the nature of your work. And you come home, maybe you have a family of five or six with three bedrooms or whatever that size of the housing could be, and you have to isolate. It becomes a challenge.*
(P030)

Further, the people who used public transportation faced challenges with respect to physically distancing themselves while on the bus, increasing their potential exposure to COVID-19. The following participant also explains how inequities in social determinants of health, such as unsafe transportation, poor housing, poverty, and food insecurity, perpetuated mistrust in the government and fueled misinformation about COVID-19.


*Folks were told to physically distance, but there were some folks who had to go to work and, you know, were being called heroes, etcetera. And so, they would be going to work early morning on the bus, but the buses were full. Folks asked for additional buses so that they could physically distance, and the official response from the government was that they weren’t going to send more buses. Oftentimes these are the communities that are living in, you know, unfit housing. They struggle wit, major issues around poverty and other issues in the community. Food security, insecurity, etcetera, across the social determinants of health. This is fertile ground, not only for distrust and misinformation for some folks.*
(P003)

#### 3.1.4. Systemic Racism

Most participants expressed how systemic racism increased the vulnerability of Black people during the COVID-19 pandemic in Canada. This includes how Black people have experienced the healthcare system and other social institutions and feel constantly left behind.


*They feel that, you know, there is nothing in the system for them. They also go back to lack of trust in the system and the medical community, and I have heard even one of our own saying that, “You know, we don’t trust you guys. We don’t trust you doctors because you just, you know, you are just a part of the whole conspiracy.” And then the historical perspective is very, very strong. People believe that, you know, nobody has their interest at heart. If not, they won’t be going through all this racism, and yet, we are asking them to be vaccinated when there are much more compelling issues around them, rather than just getting the vaccine because of getting the vaccine’s sake.*
(P024)

One participant explained Black people’s difficulties in accessing healthcare services during the COVID-19 pandemic related to anti-Black racism.


*Like for instance, we found out that during the COVID-19 spread, yeah, there have been significant experiences of discrimination among the Black people across the country. So that there were significant negative experiences in attempting to receive healthcare during the COVID-19 period.*
(P008)

Another participant felt that some Black people were refused treatment at the hospital without assessment because they had symptoms of COVID-19. For example, they describe their experience.


*I will highlight one of the situations where, you know, you just go there, before you want to seek treatment, let’s say it’s minutes before you do anything, then somebody comes to you and says, “Hey, no one can help you here. You know that there’s COVID and you have sneezed, then, you know, and you have to go home.” I mean, as much as it is for the benefit of everyone to be protected, I mean, sneezing is just a natural thing. It was there before COVID, right?*
(P020)

The following participant explains the negative impacts of healthcare racism on the African Caribbean Black (ACB) people in Canada.


*So, there is a lot of broken trust between the ACB communities and the mainstream medical healthcare because of the medical history between the ACB communities and the health system. We have a lot to reference…We could also talk about the mental health, looking at the trauma, including medical PTSDs for those who have experienced medical racism, directly or indirectly within the healthcare system. Things like paranoia and suspicion for medical interventions. And these are not all unfounded… Even recently, we still are collecting stories of racialized folks, going to the hospital, and receiving microaggressions or, getting ignored in the hospitals. A lot of ACB folks can point directly to one, or two incidents and healthcare access barriers, simply because of racism.*
(P001)

The participant further explained how many comorbidities affecting Black people in Canada were ignored, and now some individuals are experiencing worsened health conditions. She indicates the need for continued collaboration between the government, the ACB community, and other racialized communities to formulate sustainable healthcare strategies and promote reconciliation and healing.


*We have so many comorbidities that are affecting the ACB communities that have been ignored for the past two years, and we’re beginning to see the drawback of that. We’re seeing community members presenting at the hospital with Stage 4 malignancies, cancers and, you know, and disease, end stage of diseases that could have been prevented, because we have focused so much on COVID-19. We could relate this to the historical medical racism that the Black folks endure. So, what are the systems in place to ensure that the current collaborating measures continue and persist beyond the COVID to support the ACB community for other healthcare issues that have been exasperated by COVID-19? So, these are the kinds of conversations that the government needs to have with the ACB communities and other racialized communities, to let us see the sustainable plans and strategies that are being put in place to ensure that, there is total reconciliation and healing of this broken trust.*
(P001)

### 3.2. Mental Health Impacts

The COVID-19 online misinformation exacerbated mental health issues among Black people in Canada and generated fear and panic (e.g., fear and anger related to mandatory vaccine orders).


*So, I think for the people that I talked to that felt they were forced into it [vaccine], they were upset. They were upset, and they felt that their human rights were interfered with to some extent, because they were forced into making a decision that they did not feel comfortable in, and they were not ready to. So, it did affect them, and like many things, it’s in life, sometimes you—you need your job, so you will do what you have to do to continue working, so that you can.*
(P030)

Fear and anxiety about the vaccines due to misinformation penetrated community ties and caused a divide among Black community members. Some participants were concerned about a loss of community. They indicated that some community members feared being stigmatized because of accepting or declining the COVID-19 vaccine. The perceived stigma increased anxiety about COVID-19 and prevented people from getting and promoting the vaccine.


*And there’s like a lot of like there’s so many Black people who are like unvaccinated, but they just don’t want to say it to anybody, because they’re scared someone’s going to say something. But then on the other side, vaccinated people are scared to say they’re vaccinated or share their experiences, because the opposite people are like, “Oh, you’re so wrong!” Like I know, even personally, I tell people I get vaccinated, they’re like, “Oh, you’re infertile now.”*
(P023)

At some point in Canada, there were vaccine mandates for people who worked in some sectors, such as healthcare and other frontline occupations. Given that most Black people in Canada work in these sectors, the misinformation about the vaccine increased fear and worsened the mental health of Black community members who received the COVID-19 vaccine not because they wanted to but because they feared losing their jobs.


*People only got the vaccine because they were going to lose their job, right? Otherwise, they wouldn’t come to that decision themselves, without being coerced, or being forced, they’re not going to take it.*
(P012)

Further, participants reported that the COVID-19 pandemic had worsened mental health inequities in the Black communities in Canada because of the economic precarity of those who worked in frontline jobs with no other options for obtaining income.


*So, one of the biggest things I’m seeing is sort of the economic entrenchment of folk in sort of precarious work, right? So, if you have someone who, for example, is a taxi driver, right, and speaks—and is a recent immigrant, like that is—that makes a big difference between whether or not he’s going to pay the bills and support his kids, right? So of course, that person’s going to have poor mental health. You know, there’s also, on top of that, isolation, and that sort of thing. So, I think the pandemic has widened social inequality, and Black people just always have it the worst, honestly.*
(P028)

Other participants explained that public health measures, such as isolation, exacerbated the poor mental health of Black people who were already facing marginal housing with limited living spaces to maintain physical distancing and isolation.


*We know that even on just economic standpoints, a lot of the Black communities do not have the same financial resources, or even when it comes to things like housing and transportation and quality of food, and all those things that would actually give you a better starting point in life. So, when you have something that just comes and adds more burden, like if there’s a sickness—and I know the issue of isolation was a much-talked issue when it happened.*
(P030)

Despite the mental health concerns, mental health issues can be considered taboo within Black communities, and this led to limited access to and use of mental health services during the pandemic.


*It’s huge in the sense that even before the pandemic, we’ve suffered a lot of mental health issues before the pandemic... And in our community, we don’t own up to those things because we don’t think they exist. Even if you seek, you’re not going to find an answer because the system has been built in such a way that it’s not going to remove those stressors, because they are part of the structures.*
(P024)

### 3.3. Addressing Impacts of COVID-19 Pandemic

The participants understood that the impacts of the COVID-19 pandemic could not be addressed in isolation, given the disproportional effects that were exacerbated by existing health inequities and other general systemic factors. Therefore, it is crucial to understand the factors associated with the COVID-19 pandemic from an equity lens to address the needs of the Black community. The following participant explains.


*Folks are continuing to do community-based advocacy. Folks are turning, you know, the town halls into more things that respond to than just the COVID-19. Where people know the impacts of the COVID. COVID has caused educational loss. Folks are now turning their attention to how do they create the spaces where parents and/or young people themselves can get, again, the rounded out, holistic information like, you know, what might be the tools or supports to recover from education loss, or what might be the tools or resources for culturally safe mental health services that adequately address their needs? Plus, the information around COVID-19 vaccination.*
(P003)

Some participants indicated they utilized ‘Afrocentric’ approaches, i.e., collaborative interventions and culturally responsive communication, to understand community challenges and relay information about COVID-19 and vaccines.


*I’m going to highlight the ACB culture, broadly speaking, and not with any specificity. We know that many ACB cultures turn to many paths, when looking to conventional medicine. Many have their faith and beliefs that could sometimes conflict with mainstream conventional medical practices, making it important to be collaborative, you know, in our approaches and strategies in order to succeed. Honest conversation. No coercion and no judgement. No discrimination, you know, for folks who have not decided to have the vaccine, or for folks who have decided to have the vaccine.*
(P001)

Several participants highlighted how healthcare providers from Black communities engaged in initiatives that addressed COVID-19 disinformation and its impact on Black people in Canada. These included building trust with the communities, educating the communities about COVID-19 and vaccinations, debunking COVID-19 myths, and conducting targeted community testing and immunization.


*So, for example, you know, the town hall was one of them. And two, presenting ourselves in a way that people can actually see us, and also see the work that we do in our communities by engaging, by going out to educate and vaccinate. So the fact that they see you, you know, you’re next to them sitting down, talking to them, respectfully, right? That tends to work, and I’ve worked in a few vaccine clinics, you know, over the summer whereby I have seen that people trusted the Black healthcare providers.*
(P024)

Other participants indicated that some Black-led organizations worked on providing accessible information to the community, such as creating an online knowledge hub about COVID-19 for the community to easily access information about the virus and other health-related issues. Some aimed to provide digital literacy training for community members through collaboration with other Black community leaders and healthcare providers to address vaccine hesitancy and debunk myths.


*One of the things that is going on right now is that we’re trying to create digital cyber literacy for folks in the community. Because we are seeing that as much good information can be spread using digital tools, misconceptions, and information that is not verified could also be spread using digital tools. What we need to teach the community is how to know the difference in between, and how to learn to stop, not to share information when they are not sure of the sources if they are not sure that it’s verifiable. So there has been talk about religious leaders promoting this.*
(P001)

Most participants called for collaborative approaches involving the government, healthcare agencies, healthcare providers, and Black community organizations in Canada to address issues related to the COVID-19 pandemic. This includes genuine, respectful discussions of other topical issues affecting Black people, such as employment discrimination, medical racism, and anti-racist workplace practices and policies.


*The real issue for this broken trust is systemic racism. The government needs to be very intentional and honest about their responses to systemic racism, recognizing that they cannot undo the past, but they can definitely do something about it going forward. They can involve racialized groups that have been at the receiving end of this trauma, and elevate their voices, and include them in systemic approaches going forward. Getting ACB folks, ACB leaders into, you know, stable employments. ACB folks into all levels of government and all levels of governance, elevating the voices of community members, you know, of racialized communities like the Indigenous people, you know, the BIPOC [Black, Indigenous, and people of color] people.*
(P001)

Some participants talked about the capacity of building service providers to enable respectful, honest conversations about public health issues with the clients. With the understanding that service providers are also community members, the participants saw value in being intentional in building knowledge and awareness so that Black people could receive culturally appropriate care.


*I think it just comes down to healthcare professionals understanding culturally sensitive and culturally safe care and taking courses like trauma-informed care and knowing the correct way to approach things…So I think like having the training as to why people have distrust and what you can do, I think is really important.*
(P028)

## 4. Discussion

Our findings show that systemic racism in Canada exacerbated health inequities and undermined interventions meant to curb the spread and impact of the COVID-19 pandemic on Black people and other racialized communities. Participants explained how pre-pandemic inequities and a lack of tailored responses left Black Canadians largely unsupported and led to deepening health disparities.

First, this study shows how the spread of disinformation, a lack of timely and consistent information, language barriers, and structural racism left Black people uniquely and disproportionately vulnerable to the COVID-19 virus and the wider impacts of the pandemic. Participants in this study reported that disinformation increased vaccine hesitancy among Black populations in Canada at the beginning of the pandemic, echoing existing evidence that only 56.6% of Black Canadians are very or somewhat willing to receive a COVID-19 vaccine [25]. This attention to disinformation is not a new focus, as prior work has found that access to disinformation and misinformation, often shared through social media platforms, can lead to high rates of COVID-19 infection and low uptake of vaccines [20,26]. Data also shows how racialized communities may be disproportionately vulnerable to disinformation related to the pandemic because of historical and current structural inequities [27].

We advance our current understanding of the impacts of disinformation by connecting its spread to the lack of timely and accurate information and leadership. Participants explained that a lack of trusted information left space for assumptions and general misconceptions. This lack of trust in information sources can leave people without the ability to accurately gauge risk or make decisions in their best interests. Our findings show that this lack of trust in leadership and information stems from a combination of inconsistency in advice from public health agencies, a lack of culturally appropriate communication of information to Black communities, and language barriers. 

Here, our study adds to the research about public policy responses to COVID-19. Previous research has considered leadership complicity in the spread of misinformation and disinformation (especially in countries such as the United States) [27,28,29]. In Canada, these critiques have, by and large, not charged governments with malicious denialism. However, policy responses to the pandemic have been criticized, especially with respect to the lack of race-based data collection. Some have argued such a lack of data weakened public policy responses and led to greater health inequities among racialized communities [30]. 

Calls for a better understanding of racialized and intersectional pandemic impacts have led to the collection of such selected data in some places in Canada. However, the federal collection of disaggregated race-based data is still lacking. Many researchers and practitioners have noted the importance of collecting race-based data as a part of a continued policy response to the pandemic [19,31]. So far, only piecemeal demographic and racial data are available in some areas of the country and reflect the inequities found in data from the United States [1,32]. We found that the lack of disaggregated data limits public health by curtailing tailored and specific community approaches.

In explaining the impacts of mistrust in the healthcare systems, participants noted how language barriers, the lack of timely and consistent information, and the dearth of culturally appropriate approaches did not simply become a problem at the onset of the pandemic in March 2020. Instead, these failures operated in the shadow of historical (and ongoing) systemic racism within the Canadian healthcare system. Understanding the pandemic’s impacts on Black communities in Canada demands an analysis of institutionalized racism across all sectors. Participants in this study discussed how historical encounters of Black people with the Canadian healthcare system decreased vaccine confidence in ways different from White people. In the case of many Black Canadians, this hesitancy stems from fear of being experimented on or offered ineffective treatment. Racialized communities, especially Black and Indigenous communities, as well as queer people, have historically been targeted for experimentation by healthcare institutions in both the United States and Canada [30]. One recent study identified this medical legacy as a key reason for current hesitancy around novel vaccine uptake among Twitter users in Canada [33]. 

Our claims can be contextualized with the work of Krystal Batelaan [34]. Batelaan’s research from the United States shows how centuries of health disparities have devalued Black lives and resulted in vaccine hesitancy or resistance among many Black people [33]. In Canada, the focus on mistrust and vaccine hesitancy has centered on a lack of cultural competence and representation among healthcare providers [35]. Research shows that systemic anti-Black racism is prevalent in the Canadian healthcare system, impacting Black patients [36] and health professionals [37]. Our findings suggest not only a need for greater cultural training among providers and an increase in the number of Black healthcare professionals but also increased connections between communities and healthcare leaders. These connections are important to mitigate the effects of systemic healthcare marginalization and to remove barriers to health equity that are specific to the COVID-19 pandemic.

The impacts of structural racism within the healthcare system extend to health equity beyond COVID-19. Black people in Canada are more likely than White people to experience diabetes and hypertension [38], and Black women are more likely than other groups to experience other chronic health conditions such as cancer, cardiovascular disease, and depression [39,40]. Anti-Black racism has also not been isolated to the healthcare sector. It pervades all Canadian institutions and service sectors [41]. This has left Black Canadians disproportionately vulnerable to COVID-19 infection and the long-term social and economic impacts of the pandemic. 

Participants in this study also identified socio-economic impacts as a key factor for shaping the pandemic’s impacts on Black people. They reported how most people from communities in which they worked held frontline jobs. The higher risk of COVID-19 infection in Black vs. White segments of the Canadian population is driven by these higher rates of frontline work (which increases the risk of exposure to the virus) compounded by pre-existing health inequities [42]. Additionally, and as noted above, many Black people in these communities shared housing with large families, making isolation difficult or impossible. Racialized people, especially women, held disproportionately high numbers of frontline and essential service positions prior to and throughout the pandemic, especially in areas of agriculture, healthcare, and gig economy work [40]. 

Participants also noted how issues related to inequities in social determinants of health (e.g., poor housing, poverty, unsafe transportation) were related to race. Batelaan’s work is again important to consider here. She suggests a need not only to address continuing discrimination in healthcare but also for policy solutions to address the needs of Black communities beyond vaccine uptake [34]. She explains how vaccination may be a relatively low priority compared to police violence, housing, financial strain, job precarity, etc. While she speaks from the context of the United States, our study demonstrates that these problems are present in Canada as well.

A general lack of intersectional attention has been paid to the impacts of the pandemic globally. In one commentary, researchers Josephine Etowa and Ilene Hyman call for an analysis of the intersections of race, migration, and gender in exploring the impacts of COVID-19 on Black Canadians [43]. These socio-economic and health disparities have not yet been addressed and will demand systemic reform. Without such reform, issues are likely to deepen. Colliding factors of an economic “shesession” and chronic anti-Black racism have left Black Canadians, especially Black women, extremely vulnerable to marginalization from service provision, not limited in terms of the healthcare service but extending into transit and housing [40]. Our findings also show anxiety related to fear of job loss. As discussed, those who filled frontline and essential positions were left disproportionately at risk of contracting COVID-19. They were also often left in precarious positions with respect to employment as alternative workplaces and industries were shut down due to lockdown restrictions.

The second theme relates to the pandemic’s impacts on the mental health of Black Canadians. Participants explained how fear and anxiety pervaded the pandemic in some ways similar to the general population and in other ways unique to Black people. For example, participants explained how fear of mandatory vaccine orders was often shaped by historical and personal experiences of anti-Black violence within both health science and policing. 

In Canada, 26% of Black and 27% of Indigenous people reported having experienced at least one serious dispute with law enforcement, often related to discrimination, compared to 17% of those who were non-visible minorities [44]. Data on police shootings support these reports; a study by the 2018 Ontario Human Rights Commission found that between 2013 and 2017, Black people were almost 20 times more likely to be fatally shot by the police than White people [45]. One aspect of Canada’s pandemic policy response has been to massively expand policing. As police powers were widened to enable the enforcement of emergency orders and restrictions, those who have been historically and are currently vulnerable to police violence were made even more vulnerable [13,46]. News reports offer ample evidence of the dangers that came with increased policing during pandemic restrictions. In Ottawa, Obi Ifedi, a Black father, was tackled and punched by a bylaw officer and ticketed $2000 for taking his daughter to a park during lockdown restrictions, despite the fact that many other people were also in the park and were not approached by officers [47]. Parallels are evident between the fear of being forced to vaccinate and other ways in which Black people have been surveilled and regulated through the pandemic. 

Fear and anxiety also extended beyond fear of the virus and mandatory vaccination. Participants reported divides within their communities surrounding the willingness to accept COVID-19 vaccines. This often resulted in the stigmatization of one side by the other and some people choosing not to disclose their vaccination status. Though our findings did not uncover demographic trends among vaccine hesitancy, our research may deepen the understanding gained from previous quantitative studies that do break down demographic predictors. For example, one study from Ontario found that being Black, being a woman, having lower levels of income and education, and being younger were all factors that increased the likelihood of being vaccine-hesitant or resistant [48]. Taken together, these data and our results show that community divides may fall along specific demographic characteristics or have implications for some groups over others. 

Finally, participant narratives included recommendations for moving forward in addressing the impacts of the COVID-19 pandemic on Black people in Canada. Previous work has identified barriers to health equity and healthcare access that include gaps in knowledge about COVID-19 vaccines, mistrust in the government and pharmaceutical companies, fear of being experimented on, wanting others to receive the vaccine first, concern about vaccine safety and efficacy, and experiences with racism and discrimination [49]. Our work corroborates these findings but adds language barriers. Facilitators include having access to reliable information and advice from medical professionals [49]. Here, we complicate previous claims by asserting that advice from medical professionals may not actually be trusted. We instead emphasize an increased representation and cultural competency among healthcare professionals as a key condition for medical advice to be valued. We also add an emphasis on collaboration between healthcare professionals and community leaders.

Thus far, community-based approaches and Black leadership have been successful in raising levels of COVID-19 vaccine uptake among Black populations. One example of a successful initiative is the LEAPS (Listen and Learn, Empower and Engage, Ask and Acknowledge, Paraphrase and Provide, Support and Spark) of the care communication framework [50]. Another example is the Black Health Vaccine Initiative, led by the Black Physicians Association of Ontario and the TAIBU Community Health Centre, to build vaccine confidence and uptake among Black Canadians [51]. The TAIBU Community Health Centre in Ontario employs an Afrocentric approach in their cancer screening services to encourage higher rates of screening uptake among Black patients [38]. TAIBU conducts its work through core values of cooperation, the value of the collective, and a focus on dismantling systemic oppression from intersectional, equity-driven, and culturally competent service practices [39]. Participants noted the importance of Afrocentric approaches to information dissemination and care provision. 

Additionally, participants called for more accessible information. They explained how information should be provided in culturally appropriate ways, should be available in languages relevant to the communities they target, and should seek to build trust. Prior work has shown that town halls may offer a model for successful information dissemination. Town halls, which can be live-streamed over social media platforms, can use question-and-answer periods to disseminate information and dispel myths [52]. Though prior work has noted opportunities in such a model, no data yet exists on the efficacy of town halls. This could be an important direction for future research. Additionally, participants explained the importance of collaborative approaches involving governments, service providers, organizations/agencies, and communities. Here, we add to previous work [19,31,41,49,50,52] that identifies a need for greater funding for equity-building initiatives in both the United States and Canada. 

## 5. Conclusions

This study conducted a content analysis of interviews with 30 Black stakeholders across Canada. We asked stakeholders to share their thoughts on the impacts of the COVID-19 pandemic on Black people and communities in Canada. Our findings suggest that Black Canadians were made vulnerable to high rates of infection through a lack of clear and consistent information from public health leaders, the spread of disinformation, language barriers, a lack of culturally competent and tailored public policy responses, and structural racism within the Canadian healthcare sector and across all service providers. In addition to an increased vulnerability to infection and issues around vaccine hesitancy, this also left Black people in financially precarious positions and resulted in mental health implications. Moving forward, participants advised on the importance of addressing systemic inequities within and beyond healthcare; increasing cultural competency among service providers; increasing the numbers of Black healthcare professionals; increasing collaboration across sectors and between governments, service providers, organizations, and community groups; the centralization of Afrocentric approaches to healthcare; and greater capacity building for organizations engaged in equity work.

## Figures and Tables

**Table 1 ijerph-20-01580-t001:** Participant sociodemographic characteristics.

Participant #	Country of Origin	Location	Gender	Age (Years)	Religion	Fully Immunized	Role
1	Nigeria	Ottawa	Female	36	Christian	Yes	Service Provider
2	Canada	Halifax	Female	63	Christian	Yes	Community Leader
3	Canada	Toronto	Male	32	Other	Yes	Service Provider
4	Burkina Faso	Edmonton	Male	44	Christian	Yes	Community Leader
5	Nigeria	Edmonton	Male	35	Muslim	Yes	Service Provider
6	Kenya	Edmonton	Female	45	Christian	Yes	Service Provider
7	Canada	Halifax	Female	68	Christian	Yes	Service Provider
8	Nigeria	Ottawa	Male	50	Christian	Yes	Service Provider
9	Nigeria	Winnipeg	Male	54	Christian	Yes	Service Provider
10	Kenya	Ottawa	Female	40	Christian	Yes	Service Provider
11	Cameroon	Edmonton	Female	57	Christian	Yes	Service Provider
12	Rwanda	Ottawa	Male	49	Christian	Yes	Community Leader
13	Nigeria	Regina	Male	48	Christian	Yes	Community Leader
14	United Kingdom	Kingston	Male	28	Christian	Yes	Service Provider
15	Kenya	Lethbridge	Male	34	Christian	Yes	Service Provider
16	Kenya	Edmonton	Female	45–50	Christian	Yes	Service Provider
17	Ethiopia	Edmonton	Male	43	Christian	Yes	Community Leader
18	Kenya	Calgary	Male	48	Christian	Yes	Service Provider
19	Canada	Toronto	Female	44–54	Christian	Yes	Community Leader
20	Nigeria	Edmonton	Male	40	Christian	Yes	Community Leader
21	Zimbabwe	Edmonton	Female	40	Christian	Yes	Service Provider
22	Zimbabwe	Edmonton	Male	38	Christian	Yes	Community Leader
23	Nigeria	Edmonton	Male	44	Christian	Yes	Service Provider
24	Nigeria	Toronto	Female	58	Christian	Yes	Service Provider
25	Uganda	Edmonton	Male	34	Christian	Yes	Community Leader
26	Ghana	Calgary	Female	40	Christian	Yes	Service Provider
27	Sudan	Edmonton	Male	54	Muslim	Yes	Community Leader
28	USA	Vancouver	Female	29	Christian	Yes	Service Provider
29	Kenya	Ottawa	Female	57	Christian	Yes	Service Provider
30	Kenya	Ottawa	Female	53	Christian	Yes	Service Provider

## Data Availability

The data presented in this study are available on request from the corresponding author.

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
