# Peer review of "Impact of the COVID-19 Pandemic on Black Communities in Canada"

_ijerph, 2023, doi:10.3390/ijerph20021580_

Round 1
Reviewer 1 Report
It is a study that documents the impact of the covid-19 pandemic on Black communities in Canada, from the stakeholders perpective.
Lines 71-72, the objective should clearly state that the study is based on the point of view or perception of stakeholders, since reading the title one might think that it reflects the direct experience of people in these communities. The objective mentioned in the summary is clearer. I suggest ajust the title also.
Design
Line 75, “qualitative” is repeated “Qualitative descriptive qualitative research methodologies”
Authors refer a semi-structured interview questionnaire as a data collection technique, nevertheless they include two techniques, an interview and a survey (questionnaire), I think they must refer the questionnaire as an interview outline. Additionaly, interviews in-depth were mentioned in line 94, please clarify since in-depth and semi-structured interviews are different techniques, which require different application times.
Regarding the informant selection technique (line 86), in qualitative design there are many criteria that define purposive sampling processes (see for example Patton M. Qualitative Evaluation and Research Methods. Sage Publications, 1990; p. 169-183), according to the above, it seems that the authors followed a maximum variation sampling process by origin (ACB) (lines 97-99) that should be identified as such.
It is not clear if some Advisory Committee members participated also as informants, please clarify and mention how the final number of participants was defined.
In the same sense, authors should specifically mention the type of data analysis they carried out, in abstrac they mention it was a qualitative content analysis, but it is not mentioned in Data Collection and Analysis.
Results
In table 1 in the age column there are two cases (16 and 19) where there is a range, what does this mean?
Line 265 clarify what does ACB communities means, add between parenthesis African, Caribbean and Black
Line 397 clarify what does BIPOC people, add between parenthesis black, indigenous and people of color
It is important to say that the informants talk about the experience of others or how they perceive the obstacles that the Black population faced during the pandemic, all of them speak in third person, some examples of this are the following testimonies:
Like for instance, we found out that during the COVID-19 spread, yeah, there have been significant experiences of discrimination among the Black people in – in – that’s across the country. So that there was significant negative experiences in attempting to receive healthcare during the COVID-19 period. (P008)
So I think for the people that I talked to that felt they were forced into it [vaccine], they were upset. They were upset, and they felt that their human rights was interfered with to some extent,(…)
And there’s like a lot of like there’s so many Black people who are like unvaccinated, but they just don’t want to say it to anybody, because they’re scared someone’s going to say something. But then on the other side, vaccinated people are scared to say they’re vaccinated…
Rather than recovering the experience of the informants, the study recovers the point of view of members of the Black community (stakeholders) about what other Black people had to face in the pandemic. Participants do not speak from their own experience, but from what they heard, saw, and perhaps from what they interpreted, from what happened to others (perhaps) close to them. The problem is that these participants attribute meanings to what other people did, these meanings may be based on prejudices and stereotypes, and these attributed meanings are what the analysis is based on, so the result is a meta-meta-interpretation. The study has merit, but was it not possible to obtain direct experiences from Black people?
Discussion
Finally, the authors acknowledge the heterogeneity of Black communities (line 97) but do not mention any contrast in the results and discussion related to their variation criteria (ACB people). Did the analysis take this differentiation into account?
Reviewer 2 Report
I would like to thank the authors for the draft. The proposed topic seemed promising, with the dynamic changes linked to the COVID 19 pandemic presenting a case for exploration of adaptive responses by different groups within the Canadian society. In order to be publishable, the article would need to have a research question, a representative sample size, and a geographically defined area of study, as a minimum. The sample has an extremely small size. By singling out a community in Canada, it suggests a comparative angle, which is absent. Vulnerability of one group may be established in comparison to other groups, but there is no comparator. Comparators may be easy, such as the elderly, whose mortality has been cited across the world ahead of any other group within a given society? Or low income persons from other ethnic communities, or amongst Canadians? As it stands, the narrative sections of the draft read as compilation of negatively framed personal opinions, which are not supported by any verifiable evidence. In order to be published in an academic journal, they require substantiation and statistical verification, lacking from the narrative. References twist and incorrectly present ideas in the referenced sources. I randomly checked some references. The sources do not say anything of the sort implied in the draft - this needs to be adjusted.
Reviewer 3 Report
please see the attachment

Round 2
Reviewer 2 Report
I would like to thank the authors for their renewed effort. I noted the amendments to the text. Minor adjustments do not address the main concerns related to this draft. It still remains an unsubstantiated narrative of how the Black communities were misinformed and mistreated during COVID. Information coverage was equally available - and equally confusing, if you like - to all members of the Canadian society, and worldwide. How does the argument of misinformation of the Black communities hold? Nothing in the interviews you present suggests deliberate misinformation. Your interpretation of the interview findings is misleading and inaccurate (132, 144-150, etc.).
131 - evidence of systemic racism - was there a single verifiable/documented case that a member of the Black community was refused medical assistance on the basis of skin color?
"confusion" about the pandemic and vaccines was ubiquitous, and remains so, regardless of the community belonging.
Regarding the references, I now understand how the statements of the authors listed in the literature review are twisted, given the approach to the interpretation of the interviews.
